Rapid detection of enterobacteria in wastewater treated by microalgal consortia using loop-mediated isothermal amplification (LAMP)

http://orcid.org/0000-0003-4415-3108 Cameron Henry henry.cameron@uantof.cl
Bazaes Jazmín
Sepúlveda Claudia
Riquelme Carlos
Centro de Bioinnovación Antofagasta (CBIA)/Facultad de Ciencias del mar y Recursos Biológicos, Universidad de Antofagasta , Antofagasta , Chile
Neumann Bernd
Electronic publication date: 2024 Nov 15
Publication date: 2024
Volume: 12
Electronic Location ID: e18305
Received 2024 May 23; Accepted 2024 Sep 23
Copyright: © 2024 Cameron et al.
Copyright year: 2024
Copyright holder: Cameron et al.
License: This is an open access article distributed under the terms of the Creative Commons Attribution License, which permits unrestricted use, distribution, reproduction and adaptation in any medium and for any purpose provided that it is properly attributed. For attribution, the original author(s), title, publication source (PeerJ) and either DOI or URL of the article must be cited.
License URL: https://creativecommons.org/licenses/by/4.0/

Keywords: LAMP, Enterobacterias, Microalgae, Bioremediation, AMR

Funding: University of Antofagasta, Antofagasta, Chile ANID IT21I0061 This work was part of the 2023 Research Assistant project of the University of Antofagasta, Antofagasta, Chile, and the national project ANID IT21I0061 fund Implementation of the use of microalgal consortiums for wastewater bioremediation and obtaining irrigation water, a contribution to the circular economy in coastal-desert communes. The funders had no role in study design, data collection and analysis, decision to publish, or preparation of the manuscript.

==============================
In the present study, nine Enterobacteriaceae species present in wastewater were isolated and identified, and loop-mediated isothermal amplification (LAMP) was developed for the detection of Enterobacteriaceae by designing primers based on the mcr-1, KPC, OXA-23, and VIM genes, which are recognized markers of antimicrobial resistance (AMR) transmission during microalgal bioremediation treatment. The developed assays successfully detected four strains positive for mcr-1 gene-asociated resistance (Acinetobacter baylyi, Klebsiella pneumoniae, Morganella morganii, and Serratia liquefaciens), three strains for KPC gene-associated resistance (Acinetobacter sp., Escherichia coli 15499, and Morganella morganii), seven strains for OXA-23 gene-associated resistance (Acinetobacter baylyi, Enterobacter hormaechi, Enterobacter cloacae, Escherichia coli 15922, Escherichia coli 51446, Morganella morganii, and Serratia liquefaciens), and three strains for resistance to the VIM gene-associated resistance (Acinetobacter baylyi, Acinetobacter sp., and Enterobacter hormaechi) from a single colony. A reduction in microbiological load of 93.6% was achieved at 15 colony-forming units (CFU) mL−1, utilizing EMB agar and LAMP values of 0.142 ± 0.011 for the mcr-1 gene, 0.212 ± 0.02 for the KPC gene, 0.233 ± 0.006 for the OXA-23 gene, and 0.219 ± 0.035 for the VIM gene. Furthermore, bioremediation efficiency values of 71.6% and 75% for total nitrogen and phosphorus, respectively, were observed at 72 h of treatment in open pond microalgal remediation systems (MRS). This study demonstrated that the LAMP technique is faster and more sensitive than traditional detection methods, such as CFU, for Enterobacteriaceae. Consequently, this method may be considered for the detection of microbiological quality indicators within the water treatment industry.

Introduction

The treatment and reuse of wastewater have emerged as a sustainable solution to global water scarcity. This paradigm shift transforms the sector from a linear model characterized by extraction, use, and disposal to a circular model in which processes are redesigned to add value at each stage and achieve zero waste generation. Among the leading nations in wastewater reuse are the United States, Saudi Arabia, and Qatar (Jimenez & Asano, 2008).

In the field of microalgae wastewater treatment, modifications to conventional processes have been developed with the aim of treating large volumes of wastewater in a reduced time frame. These advances include techniques such as dialysis algal cultures (Marsot, Cembella & Colombo, 1991) and stabilization ponds (Kayombo et al., 2005). The utilization of biological materials as an alternative for wastewater treatment is advantageous, as it generates no secondary pollutants and is cost-effective due to its high availability and ease of procurement (Cuizano & Navarro, 2008). Microalgae have demonstrated significant potential for the removal of nitrates, phosphates, dissolved oxygen (DO), chemical oxygen demand (COD), biological oxygen demand (BOD), total organic carbon (TOC), and total dissolved solids (TDS) (Afzal et al., 2020). Sharma (2019) reported that the algal consortia selected in their study demonstrated efficient removal of nitrate, phosphate, chemical oxygen demand, ammonium, and heavy metals from wastewater samples, suggesting their potential for successful bioremediation. Nevertheless, inorganic components are not the sole elements that must be reduced in treated water, as it commonly contains various pathogenic agents, including bacteria, helminths, viruses, protozoa, and fungi (Ahmad et al., 2014). Bacteria are microorganisms primarily associated with fecal bacterial indicators of water quality, particularly Escherichia coli and certain Enterococcus species (Martzy et al., 2017). Additionally, other aerobic bacteria serve as relevant health indicators, including Pseudomonas, Acinetobacter, Klebsiella, and Aeromonas spp. (Hassan et al., 2023).

A salient feature shared by these indicator bacteria is the presence of antimicrobial resistance (AMR) genes (Hassan et al., 2023). One of the most extensively studied AMRs are carbapenemases, representing a versatile family of β-lactamases with a broad spectrum of hydrolysis on β-lactam antimicrobials. The production of carbapenemases has been recorded in Klebsiella pneumoniae carbapenemase (KPC), oxacillinase-48 (OXA-48), and Verona imipenemase (VIM), among others (Grundmann et al., 2010; Nordmann, Naas & Poirel, 2011; Wilson & Torok, 2018). Moreover, E. coli has been identified as harboring resistance to colistin, mediated by plasmid through the mcr-1 gene, which encodes a pEtN transferase (MacNair et al., 2018). These genes present potential targets for identification using molecular techniques.

The objective of this study is to develop a rapid protocol for identifying microorganisms present in wastewater treatment facilities employing a microalgal bioremediation system, utilizing the loop-mediated isothermal amplification technique (LAMP). This low-cost, sensitive, and rapid method is employed in molecular biology for the qualitative diagnosis of microorganisms of interest (Lee et al., 2019). In this context, the focus is on identifying Enterobacteriaceae harboring antimicrobial resistance (AMR) genes. Presently, the CBIA center engages in wastewater treatment and reuse activities aimed at mitigating organic loads (nitrogen and phosphate compounds) and microbiological agents to produce irrigation-quality water suitable for maintaining green areas. The development of a diagnostic tool capable of elucidating the presence of pathogens during water treatment is essential for the utilization of this treated water for irrigation purposes.

Materials and methods

Bioremediation system: The experimental setup was conducted outdoors at the pilot plant of the Bioinnovation Center of the University of Antofagasta (CBIA, UA). The household wastewater was sourced from the Autoclub treatment plant, Antofagasta (Fig. 1A). The bioremediation prototype consisted of raceways (RW) comprising two open channels through which water circulates due to the impulse generated by rotating vanes. The RWs measured 1 m in width and 3 m in length, with a capacity of 568 L at a water column height of 15 cm; thus, the prototype had a total volume of 1,700 L (Figs. 1B and 1C). Bioremediation was conducted over a 72-h, with wastewater disposed of at a ratio of 1.8:1 relative to the volume of the microalgal consortium utilized (Scenedesmus sp. and Chlorella vulgaris in a 1:1 ratio). The experiment was replicated in triplicate at an inoculated microalgal concentration of 0.41 ± 0.05 gL−1 and monitored every 24 h by recording temperature (°C), pH, and incident solar radiation (μmol·m2 s−1). For the quantification of colony-forming units (CFU) mL−1, an agar medium containing Eosin Levine Eosin Methylene Blue agar (EMBA) (211221; Becton Dickinson (BD), Pont-de-Claix, France) was employed as a selective and differential culture medium for the proliferation of coliform Gram-negative bacteria.

Figure 1 Wastewater treatment system.

(A) Autoclub treatment plant, Antofagasta; (B and C) microalgal remediation system (MRS) format in initial and final stages. Photo credit: Jaime Balbontín Westhoff and Jazmín Bazaes Donoso.

Nutrient bioremediation analysis: The concentration of ammonium was determined utilizing the Berthelot reaction (Krom, 1980). Absorbance were conducted at 690 nm on a Halo RB-10 spectrophotometer (Dynamia Scientific Ltd., Newport Pagnell, UK). Nitrate was similarly measured spectrophotometrically at wavelengths of 220 and 275 nm, adhering to the methodologies outlined by Karlsson, Karlberg & Olsson (1995). Furthermore, phosphorus quantification was performed through phosphorus-vanadate-molybdate colorimetry at 882 nm, as detailed by Rice, Bridgewater & American Public Health Association (Eds.) (2012). The nitrogen content was defined as the summation of the nitrogen contributions from both ammonium and nitrate (NH4+–N/NO3−–N). In contrast, phosphorus was calculated as the difference between the total phosphorus and the concentration of phosphate (PO43− –P).

Sample reception: Liquid samples were obtained from the pilot microalgal treatment plant (Fig. 1), contained within 250 mL sterile Schott glass bottles, and subsequently stored at 4 °C. The samples were then distributed into a series of dilution tubes and transferred from the 10−2 dilution to plates containing the differential culture medium (EMBA) using an extended seeding format. Culture plates were maintained in an incubator for 24 ± 2 h at 37 ± 1 °C, after which colony morphology and lactose fermentation capacity were assessed. Isolated colonies were subsequently transferred to new plates via streaking and cultured under the previously specified conditions. This procedure was conducted thrice until isolated colonies were achieved. Individual colonies were characterized based on criteria including shape, pigmentation, elevation, surface texture, and morphology. The resulting strains were preserved at −80 °C in cryobank beads (Copancryom).

DNA extraction: Each isolated colony obtained was subjected to DNA extraction. The colonies were inoculated in 1.5 mL Eppendorf tubes containing 200 µL of 5% Chelex buffer (Bio-Rad, Berkeley, CA, USA). Homogenization was conducted utilizing a pellet pestle mixer (Kontes Glass, Vineland, NJ, USA) for a duration of 3 min. Following this, the samples were incubated at 97 °C for 10 min using a dry bath incubator (Thermo Fisher Scientific, Waltham, MA, USA). After lysis, the pellet was subjected to centrifugation for 2 min at 7,000 rpm, allowing the recovery of DNA from the supernatant, while the pellet was discarded.

PCR: The 16S rRNA gene was amplified using polymerase chain reaction (PCR) with the following reagents: 10× buffer green, 25 mM MgCl2, 10 mM dNTP, 1 μM of each oligonucleotide, and 0.23 U μL−1 GoTaq DNA polymerase (Promega, Madison, WI, USA), employing universal primers 27F (5′-AGAGTTTGATCCTGGCTCAG-3′) and 1542R (5′-AGGAGGTGATCCAGCCGCA-3′). The PCR reaction was conducted in a thermal cycler (Veriti 96 well thermal cycler; Thermo Fisher Scientific, Waltham, MA, USA) comprising an initial cycle of 5 min at 94 °C, 30 cycles of 45 s at 94 °C, 45 s at 55 °C, and 130 s at 72 °C, concluding with a final extension of 5 min at 72 °C, and subsequent storage at 4 °C.

Sequencing and Identification: Fragment sequencing was conducted at Macrogen Inc., Seoul, Korea. The sequences were analyzed utilizing Chromas Pro and Blast software in GenBank (www.ncbi.nlm.nih.gov/blast/Blast.cgi) and the RDP Database. Sequences were aligned with Chromas Pro and compared with those available in the database.

LAMP: The LAMP assay was developed utilizing a thermostatic colorimetric sensor (MyAbscope; KANEKA, Osaka, Japan) in conjunction with the Warmstart LAMP kit (NEB, Ipswich, MA, USA). Each reaction comprised 12.5 μL of a 2× reaction mix, 2.5 μL of a primer mix, 2 μL of DNA, and nuclease-free water, resulting in a final volume of 25 μL. Primers designed by Hassan et al. (2023) were employed. The LAMP assay was executed through incubation at 65 °C for 60 min, followed by deactivation at 80 °C for 2 min, and real-time turbidity measurement at a wavelength of 575–660 nm.

Statistical analysis

A one-way analysis of variance (ANOVA) was conducted to compare the LAMP signal across various remediation times and colony formation units. Means were compared utilizing Bonferroni’s multiple comparison tests, with a p value of <0.01, employing GraphPad Prism software version 8.0.2.

Results

RWs plant treatment system and CFU: During the process of bioremediation, samples were collected for CFU analyses. Recorded values included 235 ± 27.1; 57 ± 24.5; 18 ± 6 and 15 ± 7.9 were recorded for 0, 24, 48 and 72 h following the commencement of treatment at the RW pilot plant. The samples taken at 0 hours were utilized as the control. Notably, a statistically significant difference was identified between the CFU mL −1 values of the control sample and those obtained at each subsequent hour of treatment, as denoted by asterisks (Fig. 2A). Following a 72-h exposure to microalgal treatment, the elimination rate of Enterobacteriaceae was reported at 93.6%. In the aspect of bioremediation concerning inorganic components, initial and final concentrations of 26.67 ± 2.85 and 6.24 ± 1.29 mg L−1 for nitrates and 8.86 ± 0.006 and 2.1 ± 0.10 mg L−1 for phosphates were recorded, achieving bioremediation efficiency of 71.6% and 75% for nitrates and phosphates, respectively, after 72 h of treatment (Figs. 2B and 2C). The microalgal consortium achieved a bioremediation efficiency of approximately 72% after 72 h of treatment for total nitrogen and phosphorus (Fig. 2C). Notably, reductions in nitrogen concentration were observed, decreasing from 26.4 to 6.3 mg L−1, and for phosphorus, from 8.1 to 2.2 mg L−1. Regarding the operating parameters of bioremediation, the average recorded temperature was 20 ± 2.3 °C, the pH peaked at 8 ± 0.10, and incident radiation was measured at 710 ± 130 μmol·m2 s−1. Conversely, an alkalization trend in the RW system was noted, with values transitioning from 7.26 ± 0.38 to 8.0 ± 0.10, alongside a temperature fluctuation ranging from 23.5 ± 0.19 and 18.7 ± 0.1 °C (Fig. 2D).

Figure 2 Physical/chemical and biological bioremediation assays.

Performance of microalgae in bioremediation systems for 72 h. (A) Colony forming units (CFU mL−1) during the bioremediation treatment time in RWs system. The asterisks indicate significant differences between the control (0 h) and the different treatment hours, with a confidence level of 95% after a Bonferroni’s correction, for n = 12, at p < 0.01; (B) nitrogen concentration (N–NH4 + N –NO3), phosphorus concentration (P–PO4); (C) nutrient bioremediation efficiency of nitrates and phosphates; (D) temperature and pH recording in the bioremeasurement system.

Isolation and identification of Enterobacteriaceae from wastewater: Through the examination of wastewater sourced from the Antofagasta autoclub treatment plant (Fig. 1A) enterobacteriaceae bacteria were meticulously isolated and identified. Following their isolation on eosin methylene blue (EMB) agar plates, a total of nine distinct bacterial strains were cultivated, and subsequent sequencing of the 16S rRNA genes from each isolate was performed. Notably, a comparison of the resultant sequences with the NCBI database revealed a significant likeness, with all sequences exhibiting over 90% identity, 92% query coverage, and an E value of 0. The taxonomy of the isolated bacteria was determined, indicating their classification as members of the Acinetobacter, Enterobacter, Escherichia, Klebsiella, Morganella, and Serratia genera, as summarized in Table 1.

Table 1 Taxonomic classification of nine enterobacteria isolated from wastewater.

GenBank code	Genus	Identity (%)	Query coverage	Evalue	
KY817316.1	Acinetobacter baylyi	95.79	100	0	
MH185836.1	Acinetobacter sp.	97.44	99	0	
OR889414.1	Enterobacter hormaechi	100	100	0	
MN208123.1	Enterobacter cloacae	98.03	99	0	
OR133169.1	Escherichia coli (15922)	100	100	0	
CP127316.1	Escherichia coli (51446)	100	100	0	
OQ550178.1	Klebsiella pneumoniae	96.51	95	0	
OR121927.1	Morganella morganii	100	100	0	
MH001385.1	Serratia liquefaciens	91.6	92	0	

Specificity and sensitivity of the LAMP assay: Concerning the specificity for the comprehensive detection of enterobacteria, the developed assays effectively identified four strains exhibiting resistance to the mcr-1 gene (Acinetobacter baylyi, Klebsiella pneumoniae, Morganella morganii, and Serratia liquefaciens), three strains exhibiting resistance to the KPC gene (Acinetobacter sp., Escherichia coli 15499, and Morganella morganii), seven strains exhibiting resistance to the OXA-23 gene (Acinetobacter baylyi, Enterobacter hormaechi, Enterobacter cloacae, Escherichia coli 15922, Escherichia coli 51446, Morganella morganii, and Serratia liquefaciens), and three strains exhibiting resistance to the VIM gene (Acinetobacter baylyi, Acinetobacter sp., and Enterobacter hormaechi) (Table 2). Additionally, in this study, the sensitivity of all Enterobacteriaceae strains was evaluated in each assay for the different resistance genes, achieving 1 CFU per reaction, employing the DNA extraction method (Fig. 3).

Table 2 LAMP for AMR genes.

Bacterial strains					
	mcr-1	KPC	OXA-23	VIM	
Acinetobacter baylyi	+	−	+	+	
Acinetobacter sp.	−	+	−	+	
Enterobacter hormaechi	−	−	+	+	
Enterobacter cloacae	−	−	+	−	
Escherichia coli (15922)	−	+	+	−	
Escherichia coli (51446)	−	−	+	−	
Klebsiella pneumoniae	+	−	−	−	
Morganella morganii	+	+	+	−	
Serratia liquefaciens	+	−	+	−	
Note:

Positive and negative LAMP reaction for AMR antibiotic resistance genes from enterobacterias strains. (+ positive signal; − negative signal).

Figure 3 LAMP assays from single CFU.

LAMP assay for the detection of Enterobacteriaceae isolated from wastewater, recorded using a thermostatic sensor (KANEKA, Osaka, Japan) for 60 min of testing. Positive signal for AMR, (A) MCR-1; (B) KPC; (C) OXA-23 and (D) VIM, from 1 CFU for Acinetobacter baylyi, Acinetobacter sp. strains; Enterobacter hormaechi; Enterobacter cloacae; Escherichia coli 15499; Escherichia coli 51446; Klebsiella pneumoniae; Morganella morganii; Serratia liquefaciens; and negative control (CN).

LAMP trial on bioremediation system: The PCR product’s positive signal were detected through the LAMP assay over 0, 24, 48, and 72-h intervals of wastewater remediation treatment using a microalgal consortium comprising Scenedesmus sp. and Chlorella vulgaris (Table 3). Notably, a reduction in signal intensity was observed following 72 h of the remediation process. The recorded values for the mcr-1 gene, KPC gene, OXA-23 gene, and VIM gene were 0.142 ± 0.011, 0.212 ± 0.02, 0.233 ± 0.006, and 0.219 ± 0.035 (Fig. 4) respectively. A statistically significant difference was solely identified in the signal of the mcr-1 gene.

Table 3 LAMP signal on bioremediation system.

Record of LAMP values on remediation treatment for 72 h in raceway with microalgal consortium.

	Time (hrs)	AMR	LAMP	
Time (min)	
1	10	20	30	40	50	60	
Bioremediation treatment time (hrs)	0	mcr-1	0.001 ± 0.003	0.015 ± 0.008	0.021 ± 0.008	0.124 ± 0.069	0.196 ± 0.005	0.212 ± 0.013	0.212 ± 0.013	
		KPC	0.001 ± 0.001	0.004 ± 0.002	0.014 ± 0.005	0.139 ± 0.020	0.181 ± 0.005	0.203 ± 0.008	0.217 ± 0.009	
		OXA-23	0.001 ± 0.001	0.005 ± 0.001	0.008 ± 0.001	0.040 ± 0.035	0.165 ± 0.045	0.235 ± 0.044	0.262 ± 0.056	
		VIM	0.001 ± 0.001	0.006 ± 0.001	0.008 ± 0.001	0.029 ± 0.001	0.171 ± 0.034	0.213 ± 0.056	0.257 ± 0.068	
	24	mcr-1	0.004 ± 0.003	0.004 ± 0.003	0.008 ± 0.001	0.081 ± 0.066	0.155 ± 0.031	0.177 ± 0.033	0.187 ± 0.035	
		KPC	0.001 ± 0.005	0.013 ± 0.003	0.019 ± 0.001	0.120 ± 0.015	0.170 ± 0.030	0.199 ± 0.005	0.212 ± 0.030	
		OXA-23	0.001 ± 0.005	0.011 ± 0.003	0.015 ± 0.001	0.20 ± 0.006	0.253 ± 0.007	0.278 ± 0.005	0.293 ± 0.013	
		VIM	0.001 ± 0.004	0.016 ± 0.005	0.021 ± 0.005	0.038 ± 0.023	0.127 ± 0.093	0.207 ± 0.014	0.246 ± 0.015	
	48	mcr-1	0.012 ± 0.004	0.012 ± 0.004	0.010 ± 0.002	0.127 ± 0.028	0.160 ± 0.031	0.178 ± 0.036	0.189 ± 0.039	
		KPC	0.001 ± 0.005	0.014 ± 0.003	0.018 ± 0.003	0.125 ± 0.004	0.161 ± 0.086	0.202 ± 0.109	0.222 ± 0.108	
		OXA-23	0.001 ± 0.002	0.011 ± 0.002	0.016 ± 0.001	0.180 ± 0.036	0.242 ± 0.016	0.258 ± 0.026	0.269 ± 0.034	
		VIM	0.001 ± 0.001	0.013 ± 0.003	0.017 ± 0.004	0.129 ± 0.047	0.212 ± 0.020	0.245 ± 0.013	0.267 ± 0.013	
	72	mcr-1	0.015 ± 0.007	0.015 ± 0.007	0.013 ± 0.002	0.020 ± 0.002	0.115 ± 0.001	0.134 ± 0.007	0.142 ± 0.011	
		KPC	0.001 ± 0.005	0.010 ± 0.007	0.015 ± 0.009	0.072 ± 0.039	0.162 ± 0.009	0.194 ± 0.018	0.212 ± 0.022	
		OXA-23	0.001 ± 0.003	0.006 ± 0.003	0.009 ± 0.003	0.165 ± 0.009	0.225 ± 0.007	0.233 ± 0.006	0.233 ± 0.006	
		VIM	0.001 ± 0.001	0.012 ± 0.007	0.016 ± 0.009	0.042 ± 0.021	0.139 ± 0.045	0.193 ± 0.015	0.219 ± 0.035	
	Control	mcr-1	0.001 ± 0.002	0.014 ± 0.001	0.017 ± 0.001	0.019 ± 0.001	0.019 ± 0.024	0.019 ± 0.075	0.024 ± 0.075	
		KPC	0.001 ± 0.005	0.017 ± 0.001	0.023 ± 0.002	0.026 ± 0.002	0.027 ± 0.002	0.075 ± 0.084	0.089 ± 0.108	
		OXA-23	0.005 ± 0.002	0.010 ± 0.004	0.015 ± 0.003	0.017 ± 0.003	0.018 ± 0.002	0.018 ± 0.002	0.018 ± 0.002	
		VIM	0.001 ± 0.001	0.014 ± 0.001	0.017 ± 0.001	0.019 ± 0.001	0.019 ± 0.001	0.019 ± 0.001	0.137 ± 0.007	
Note:

AMR, Marker antimicrobial resistance; LAMP, loop-metiated isothermal amplification.

Figure 4 LAMP assays on bioremediation system.

Turbidity values of the LAMP assay amplification for the genes MCR-1, KPC, OXA-23, VIM with a confidence level of 95% after a Bonferroni’s correction, for n = 24, at p < 0.01. (* = significant difference; ns = not significant difference).

Discussion

Microalgal systems have been regarded as a viable alternative for wastewater disinfection (Ji, 2022; Jiang et al., 2018; Wollmann et al., 2019; Yong et al., 2021). It has been observed that the cultivation of microalgae in wastewater creates adverse conditions for numerous pathogens, primarily due to operational characteristics (Muñoz & Guieysse, 2006; Ruas et al., 2018). Additionally, certain physiological characteristics of microalgal systems may significantly contribute to pathogen elimination, including (i) the production of antimicrobial metabolites and toxins; (ii) the capacity to bind to bacteria and facilitate sedimentation (Liu, Hall & Champagne, 2018). The present study investigates the bioremediation of household wastewater utilizing microalgal consortia over a period of 72 h, resulting in reductions of nitrogen and phosphorus content. One of condition examined that affects algal activity on pathogens is the role of carbonate and bicarbonate ions, which react to increase carbon dioxide available to the algae while producing an excess of hydroxyl ions. Consequently, the pH of the water may rise above 9, thereby potentially eliminating fecal coliforms (Abdel-Raouf, Al-Homaidan & Ibraheem, 2012). In this study, an increase in pH was observed within the bioremediation system, achieving values of 8 ± 0.10 during the treatment period. While pH values exceeding 9 have been recorded in the RW system, this was associated with a longer bioremediation duration; however, to ensure competitiveness with conventional wastewater treatment systems (aerobic, anaerobic, and biofilm remediation methodologies), the treatment duration in RW systems must be <72 h, which influences the remediation time in microalgal treatment facilities (Sangamnere et al., 2023). Furthermore, the reduction of coliforms has been documented since 1968, when Moawad (1968) demonstrated that environmental conditions conducive to algal growth are detrimental to coliform survival. In this context, the microalga Chlorella vulgaris achieved a total elimination of coliforms, ranging from 88.7% to 99.4% (Ruas et al., 2018). Ahmad et al. (2014) noted complete elimination of fecal coliforms utilizing the algae Rhizoclonium implexum. Similarly, Amaro et al. (2023) reported total elimination rates of coliforms and Escherichia coli at 99.4% and 98.6%, respectively. Consequently, microalgal systems are recognized as a promising approach for developing novel strategies and methodologies for wastewater disinfection through the reduction of pathogens, including fecal coliforms (Bouki, Venieri & Diamadopoulos, 2013; Ruas et al., 2022).

This study demonstrates a 93.6% reduction in the population of Enterobacteriaceae following a 72-h bioremediation treatment, as assessed by CFU on EMB agar. Although CFU detection is a reliable method, it is characterized by a time-intensive process, necessitating a waiting period of at least 24 h for result interpretation. Alternative coliform detection methodologies have been developed, utilizing specific enzymatic activities for the analysis of water samples; however, these methods are associated with high costs and extended analysis times (Maheux et al., 2015).

In contrast, the Loop-mediated isothermal amplification technique, introduced by Notomi et al. (2000), offers more rapid and straightforward operations. Significant research by Hassan et al. (2023) advocates for the application of LAMP in the identification of antibiotic-resistant microorganisms, such as E. coli, Klebsiella, Salmonella enterica, and Acinetobacter, in environmental water samples. Therefore, for the assessment of microbiological water quality post-treatment, bacterial counts remain below the permissible threshold. The CFU method serves as the conventional approach for estimating the density of viable bacterial cells; however, the LAMP technique detects DNA quantities in reaction tubes rather than bacterial cells. This study integrates both methodologies to develop a detection tool for Enterobacteriaceae within wastewater bioremediation systems, with particular emphasis on the detection of antimicrobial resistance genes.

The extensive distribution of beta-lactamases, particularly carbapenemases, is a significant concern in the field of microbiology. These enzymes are frequently associated with mobile genetic elements, which facilitate their transference among bacterial species and exacerbate their relevance to public health (Hiller et al., 2019). The gene blaKPC imparts resistance to carbapenems (Codjoe & Donkor, 2018; Stoesser et al., 2017). The dissemination of blaKPC has become geographically extensive, with detection in various bacterial species, including E. coli, E. cloacae, and Serratia marcescens (Bryant et al., 2013; Stoesser et al., 2017). In contrast, OXA beta-lactamases, including OXA-48 and OXA-23, are classified as class D carbapenemases due to their rapid mutation capabilities (Codjoe & Donkor, 2018; Pitout et al., 2019). Additionally, the gene blaVIM is categorized within class B carbapenemases and is commonly found on mobile genetic elements situated within integrons (Codjoe & Donkor, 2018; Sedighi et al., 2014). Additionally, the misuse of colistin, a critical last-resort antibiotic for Gram-negative bacteria, in food production has led to an upsurge in plasmid-mediated resistance to colistin (Al-Tawfiq, Laxminarayan & Mendelson, 2017). Consequently, the transmission genes present in contaminated water represent promising candidates for targeted genetic studies.

This study successfully identified seven cultivable strains of Enterobacteriaceae using LAMP technique targeting the OXA-23 gene. Notably, two bacterial strains, specifically Klebsiella pneumoniae and Acinetobacter sp., remained undetected among the total strains employed as indicators of microbiological quality in treated water. Nonetheless, through the application of the mcr-1 and VIM genes, these previously undetected strains were successfully identified. Consequently, the comprehensive utilization of a gene pool associated with AMR demonstrates promising potential in facilitating the detection of all cultivable bacterial strains. Tongphrom et al. (2018) emphasize that LAMP assays exhibit tenfold greater sensitivity in comparison to polymerase chain reaction (PCR) methods. The enhanced sensitivity observed in this study not only corroborates this assertion but also underscores the technique’s efficacy in detecting bacterial presence at a detection threshold of merely one CFU per assay.

Conclusion

Enterobacteriaceae serves as natural indicators and are employed in the assessment of the microbiological quality of treated wastewater for various reuse applications. The efficacy of wastewater disinfection is typically gauged by the degree of removal of total coliform organisms and the reduction of inorganic compounds. This study documented a microbiological reduction of 93% for total nitrogen and 75% for total phosphorus in the MRS pilot plant after 72 h of bioremediation. Additionally, a rapid and cost-effective detection method is established through LAMP assays, which include four AMR (mcr-1, KPC, OXA-23, VIM). The research successfully identified Enterobacteriaceae within a 60-min LAMP assay. Integrated detection of AMR using LAMP in treated water samples from the RWs pilot plant revealed the presence of 1 CFU mL−1 of enterobacteria, which corresponds to the minimum microbiological unit permissible under regulatory standards. Consequently, the development of a pathogen detection technique of greater significance within a coliform water treatment facility is warranted to produce a product that complies with existing regulations. As indicated, the reuse of industrial wastewater represents an environmentally beneficial alternative to mitigate water shortages in depleted regions and address the challenges associated with the misuse of industrial wastewater.

Supplemental Information

Supplemental Information 1 Colony-forming unit related to microalgae bioremediation treatment.

Raw data for Figure 2A.

Supplemental Information 2 Nitrogen concentration (N-NH4+ N-NO3) and Phosphorus concentration (P-PO4) related to microalgae bioremediation treatment.

Raw data for Figure 2B.

Supplemental Information 3 Biorremediation efficiency related to microalgae bioremediation treatment.

Raw data for Figure 2C.

Supplemental Information 4 Temperature and pH recording in the bioremeasurement system.

Raw data for Figure 2D.

Supplemental Information 5 MCR_AMR gene.

Raw data for Figure 3A.

Supplemental Information 6 KPC_AMR gene.

Raw data for Figure 3B.

Supplemental Information 7 OXA-23 AMR gene.

Raw data for Figure 3C.

Supplemental Information 8 VIM_AMR gene.

Raw data for Figure 3D.

Supplemental Information 9 Turbidity values of the LAMP assay amplification for the genes mcr-1, KPC, OXA-23, VIM on biorremediation system.

Raw data for Figure 4.

Supplemental Information 10 Taxonomic classification of nine enterobacteria isolated from wastewater.

Supplemental Information 11 Positive and negative LAMP reaction for AMR antibiotic resistance genes from enterobacterias strains.

Supplemental Information 12 Record of LAMP values on remediation treatment for 72 hours in raceway with microalgal consortium.

Additional Information and Declarations

Competing Interests

Author Contributions

Data Availability

The authors declare that they have no competing interests.

Henry Cameron conceived and designed the experiments, performed the experiments, analyzed the data, prepared figures and/or tables, authored or reviewed drafts of the article, and approved the final draft.

Jazmín Bazaes analyzed the data, authored or reviewed drafts of the article, and approved the final draft.

Claudia Sepúlveda analyzed the data, authored or reviewed drafts of the article, and approved the final draft.

Carlos Riquelme analyzed the data, authored or reviewed drafts of the article, and approved the final draft.

The following information was supplied regarding data availability:

The raw data for Figures 2, 3 and 4 are available in the Supplemental Files.

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
