# Peer review of "Rapid detection of enterobacteria in wastewater treated by microalgal consortia using loop-mediated isothermal amplification (LAMP)"

_PeerJ, doi:10.7717/peerj.18305_

## Round 0.1 · original submission · Major Revisions

· Academic Editor

Major Revisions

The submitted manuscript was positively evaluated in general, but with several issues that need to be reviewed. One important point is the language, which uses several non-conventional wording and phrases for a scientific manuscript. Possibly a language editing service would be of benefit for the text. The reviewers opened questions regarding experimental design and reproducibility. These should be the second major focus of the revision. The editorial decision is based on these points: major revisions.

Reviewer 1 ·

Basic reporting

- line 21: please correct in listing of bacteria by replacing "y" (Morganella morganii "y" Serratia liquefaciens)
- line 50: please write "Escherichia coli" in italic letters
- line 83: please explain tjat EMB agar is used for selection of Enterobacteriaceae
- line 139: why are you starting with figure C? Please change your arrangement in figure 2.
- line 143/144: please rephrase sentence "Where CFU mL-1 values of 235 ± 27.1; 57 ± 24.5; 18 ± 6; and 15 ± 7.9 were reached for 0, 24,144 48, and 72 hours of treatment at the RW pilot plant" in more professional English.
- line 154: Please form a professional English sentence instead of writing "From the isolation in the EMB plates".
- line 172: Highlighting the decrease in signal after 72 hours of remediation treatments. Obtaining values of 0.142 ± 0.011 for the mcr-1 gene, 0.212 ± 0.02 for the KPC gene, 0.233 ± 0.006 for the OXA-23 gene, and 0.219 ± 0.035 for the VIM gene"
-line 189-194: this sentence is too long and has to be rephrased
-line 205: please rephrase: "...at least 24 hours until results are available"
- line 211: at which value is the bacterial cound below the permissible limit? Please clarify.
- line 218: please explain blaKPC
- line 220-22: this sentence is not understandable. Please rephrase.
- line 224: please use another word for "furthermore" as this was written in the sentence before.
- line 246: "Highlighting the sensitivity and speed of the technique" This sentence is not in context of the sourrunding sentences. Please rephrase.

- the authors should discuss whether you can use LAMP also for the detection of other carbapenemases like NDM

Experimental design

- for me, it is still unclear whether LAMP can be used to detect both the bacterial species and the carbapenemases. Or do you have to use the agar plates for species identification. As I undersand, you used both techniques in parralel. Where is the benefit if the time consuming part of agar plates have to be used in addition? Please clarify.

Validity of the findings

Please state the benefit of the findings in more detail.

Reviewer 2 ·

Basic reporting

- The work is focused on LAMP reactions to detect enterobacteria in wastewater. However, the introduction lacks a review of other emerging technologies, such as sensors and biosensors for genetic amplification. Also, the state-of-the-art shows some sensing devices aided with AI and learning methods to provide automated results in amplification reactions. The authors must consider adding some of the following references in the manuscript's introduction.

E.g.
https://doi.org/10.1016/j.cclet.2023.109220
https://doi.org/10.3390/chemosensors11040230

- There are some grammatical errors and formatting mistakes. For instance, the abstract (line 26) uses the acronym RW; however, it does not make sense with remediation systems. Also, the acronym CFU is defined twice in the abstract. The authors should thoroughly revise the manuscript.


- The abstract, introduction, and conclusion should be clearer about the paper's main contribution.

Experimental design

- Regarding LAMP assays, the statement in line 128 is confusing as it talks about turbidity measurement. However, in the lines above, fluorescence detection is mentioned. So, which transduction mechanism was used? If turbidity, how was the operating wavelength selected? Why a wide spectral range of 575-660 nm? This latter may contradict real-time measurements. This section should be improved.


- In Figure 3, how was the “turbidity” computed? Is it a relative or absolute measure? Be aware of the units.

Validity of the findings

- The conclusion should clarify the paper's main contribution.
- Results of comparing LAMP with PCR are missing.

Additional comments

No comment

Reviewer 3 ·

Basic reporting

Language:

At the abstract, “…VIM gene (Acinetobacter baylyi, Acinetobacter sp., Enterobacter hormaechei), from a colony-forming unit (CFU).” Normally, CFU refers to the numerical unit (number of bacterial colonies per ml) per se, therefore it is recommended to use a single colony or one colony. Plus, please double-check if any words were abbreviated multiple times. Is CFU colony-forming unit or colony formation unit?

Minor comments:

Some sentences are a bit wordy to be considered scientific writing. For example, line 52 can be revised into a simpler form with a clearer structure, such as “The shared characteristics of these health indicator bacteria…”; Lines 94-95: Samples collected from the RW were stored in 250 mL bottles at 4°C; etc.

Some words need to be refined to increase readability - Line 73: `corresponds to` should be `was`; `a water column height` can be just `height`; Line 91: `although phosphorus` can be `The phosphorus`; `expressed as the subtraction of` should be `calculated from phosphate`; `a battery of serial dilution` should be `a series of serial dilution tubes`; `10-2` is missing a superscript; lines 104, 114: will be` should be `were` (the same error can be found many times throughout the manuscript); line 138: `used achieved` should be just `achieved`, and many more.

Experimental design

As the experimental design is extremely simple, there are no identifiable flaws in the design section.

Validity of the findings

Novelty: The abstract of this manuscript insists that LAMP was developed in this study, but apparently, the primers were designed in an earlier study and this study employed the primers from Hassan et al. (line 126). Although the PEERJ journal does not require novelty of the work submitted, the authors should disclose which part of the study was actually conducted in this study.

Reproducibility: As the major finding in this study is the result of the LAMP reaction on wastewater treated by microalgae, the result should be reproducible.

---

## Round 0.2 · Minor Revisions

· Academic Editor

Minor Revisions

The reviewers requested some minor points, mainly based on language and grammar. Please consider these suggestions and re-submit after minor revision.

Reviewer 1 ·

Basic reporting

- Line 54: please write “AMRs are carbepenemases” instead of “is”
- Line 103 + 106 + 114 + 126: please add the headquarter of the companies
- Line 122+123: please use the past tense in this sentence
- Line 126-127: please add a space between number and unit
- Line 156+157: Please remove dot after bracket that the sentence is understandable
- Line 178/180: please cite Figure 4 one sentence earlier as there are already numbers arising that were shown in Figure 4.

Experimental design

no comment

Validity of the findings

no comment

Additional comments

no comment

Reviewer 3 ·

Basic reporting

Clear and unambiguous, professional English used throughout:

The authors have revised the manuscript; however, some minor points should be addressed before publication.

Throughout the manuscript: please note that the HGNC endorses the use of italics to denote genes, alleles and RNAs to distinguish them from proteins. Please refer CDC's guidelines for nomenclatures as well. https://wwwnc.cdc.gov/eid/page/scientific-nomenclature#:~:text=Bacteria%20gene%20names%20are%20always%20written%20in%20italics.&text=Fungus%20gene%20names%20are%20generally,terminology%20allows%20for%20a%20superscript.

Grammatical errors and typos: Currently the manuscript has many grammatical errors (mostly about tense and plural/singular words) and typos.

Abstract:
Phrases like "strains for KPC gene resistance", "mcr-1 gene resistance" - this may confuse potential readers, suggesting that the LMAP is detecting regions resistant to the gene rather than to the antibiotics. I suggest using "positive for mcr-1 gene", "KPC gene", ... etc. Else, "KPC gene-associated resistance" can be used.

Introduction:
Line 52: "...have emerged as a sustainable",
Line 61: Either one of "...high availability and ease of procurement" or "...high availability and ease of recovery" based on the author's context
Line 66: "successful bioremediation".
Line 89: typo at "presence of".

Methods:
Line 322: "Absorbance readings were"
Line 336: "..was repeated a minimum of three times until .."
Line 342: "pellet pestle" ?
Line 344: typo at "underwent"
Line 351: typo, missing space at "followed by "
Line 672: "Alignments were executed"...however I would rather use "Sequences were aligned with Chromas Pro and compared with those ..."
Line 677: typo at "in conjunction with.."

Results:

Line 717: it seems the original manuscript is containing the letter "o" instead of the number "0"
Line 733: typo in "Enteorbacteriaceae", it should be "Enterobacteriaceae"
Line 735: form -> from; Notable -> Notably
Line 751: The PCR products positive signals was --> "The PCR products' positive signals were" or "The PCR product positive signals were "

Discussion:
Line 985: use "pathogen elimination" or "elimination of pathogens"
Line 988: use "one condition" or "key conditions"

The sentence "One examined conditions affecting algal activity on pathogens is the role of carbonate and bicarbonate ions, which react to yield increased carbon dioxide to the algae while producing an excess of hydroxyl ions." is hard to understand. Consider rephrasing to “One of the conditions examined that affects algal activity on pathogens is the role of carbonate and bicarbonate ions, which react to increase carbon dioxide available to the algae while producing an excess of hydroxyl ions.”

nonetheles --> nonetheless, upsurge --> upsurge, genecti --> genetic, detecticion --> detection, ...and so on.

Please review the manuscript for typos before online publication.

Experimental design

There is no experimental flaws and the authors have elucidated the concept in the response to the reviewers.

Validity of the findings

The findings of this study appear to be reproducible. The authors indicated that they are using these methods extensively for further experiments, as noted in their response to the reviewers. Therefore, the findings are valid.

Additional comments

-

---

## Round 0.3 · accepted · Accept

· Academic Editor

Accept

Dear authors, I am very pleased to inform you, that your revised manuscript is accepted.